# Gell: A GPU-powered 3D hybrid simulator for large-scale multicellular system

**Jiayi Du**[1], **Yu Zhou**[2], **Lihua Jin**[2], **Ke Sheng**[3]*

**1** Department of Radiation Oncology, University of California, Los Angeles, Los Angeles, California, United States of America, **2** Department of Mechanical and Aerospace Engineering, University of California, Los Angeles, Los Angeles, California, United States of America, **3** Department of Radiation Oncology, University of California, San Francisco, San Francisco, California, United States of America

* Ke.Sheng@ucsf.edu

**Data Availability Statement:** All source code and example data are available on GitHub at https://github.com/PhantomOtter/Gell. Generated HDS simulation results have been uploaded to the

## Abstract

As a powerful but computationally intensive method, hybrid computational models study the dynamics of multicellular systems by evolving discrete cells in reacting and diffusing extracellular microenvironments. As the scale and complexity of studied biological systems continuously increase, the exploding computational cost starts to limit large-scale cell-based simulations. To facilitate the large-scale hybrid computational simulation and make it feasible on easily accessible computational devices, we develop Gell (GPU Cell), a fast and memory-efficient open-source GPU-based hybrid computational modeling platform for large-scale system modeling. We fully parallelize the simulations on GPU for high computational efficiency and propose a novel voxel sorting method to further accelerate the modeling of massive cell-cell mechanical interaction with negligible additional memory footprint. As a result, Gell efficiently handles simulations involving tens of millions of cells on a personal computer. We compare the performance of Gell with a state-of-the-art paralleled CPU-based simulator on a hanging droplet spheroid growth task and further demonstrate Gell with a ductal carcinoma *in situ* (DCIS) simulation. Gell affords ~150X acceleration over the paralleled CPU method with one-tenth of the memory requirement.

## Introduction

Computational modeling has become an important tool for studying the dynamics of tissue development and tumor response to different therapeutic interventions over the past three decades. Three major types of models are commonly utilized in these studies: discrete, continuum, and hybrid models [1]. Discrete models, also known as cell-based models or agent-based models, simulate the individual behaviors and the mutual interactions of the cells in a system. Continuum models consider biological tissues as domains composed of different solid and fluid phases and describe the system evolution using partial differential equations. Hybrid models combine the aforementioned two methods as they model discrete cells in a continuum environment.

Cell-based and hybrid simulations adopt the discrete representation of the cell of interest. Compared to the continuum representation, the discrete modeling of individual cells better captures the heterogeneity of tumorous tissue with independently tracked cell states and

github repository. https://github.com/PhantomOtter/Gell/tree/main/data.

**Funding:** The research is supported in part by NIH R01CA255432.

**Competing interests:** The authors have declared that no competing interests exist.

enables a more straightforward translation from biological hypothesis to simulation rules [2]. Cell-based/hybrid simulation has been utilized to explore various kinds of oncological topics [3], such as epithelial ducts and cysts through the epithelial acini models [4, 5], the development of the initial phases of avascular tumor development through multicellular tumor spheroid (MCS) models [6], angiogenesis, vascular network formation problems [7], and anti-cancer treatments effectiveness modeling [8, 9]. The method for discrete representation of cells can either be lattice-based or off-lattice. Lattice-based methods are mesh-based, including cellular automaton (CA) models [10], lattice gas CA (LGCA) models [11], and Cellular Potts models (CPM) [12]. Lattice-based models are susceptible to grid biases [13], which do not affect the off-lattice methods. Two major types of off-lattice models are Boundary-tracking models [14] and Center-Based models (CBMs) [1, 15]. Boundary-tracking models dedicate more computational resources to the cells' morphological dimension and are thus more computationally expensive than CBMs [2]. CBMs assume a spherical cell shape and represent cell movement by displacing the cell center position. Cell movements can be realistically modeled by incorporating forces such as adhesive, repulsive, locomotive, and drag-like forces [15]. Center-based representation is often a superior choice for large-scale cell-based/hybrid simulations interested in the heterogeneous development of biological tissue due to its realistic modeling of multicellular interactions and lower computational cost than other cell-based methods.

However, even with a center-based representation of discrete cells, as the scale and complexity of the studied biological systems continuously increase, the exploding computational cost still limits the large-scale cell-based/hybrid simulations. Several CPU-based cell-based/hybrid simulation software frameworks have been proposed to enable large-scale biological system modeling on high-performance computers. Biocellion [16], a closed-source commercial software, has simulated millions to billions of cells on a computer cluster. PhysiCell [15] is an open-source parallel simulation platform capable of simulating 18.2 days of hanging drop spheroid growth with up to one million cells in 3 days on a high-performance computer. BioDynaMo [17] is another open-source parallel simulator that is 945X faster on the 'epidemiology (medium-scale)' benchmark using 72 CPUs on a server compared to a single thread version. Because of the demonstrated potential to accelerate cell-based simulation with parallelization, recent research has shifted to the graphic processing unit (GPU), which has an intrinsic parallel architecture with thousands of computational cores. Ya||a [18] is a paralleled agent-based model on GPU. Its extended spheroid cell model with spin-like polarities can simulate epithelial sheets and tissue polarity. Although Ya||a [18] can achieve 10X acceleration compared to CPU-based cell-based simulation library Chaste [19], their simulation software is not designed for large systems. Simulation of large systems using Ya||a [18] is limited by its computational complexity of $O(N^2)$ for cell-cell interaction. CBMOS [20] is another GPU-based software that provides a platform to study the effects of force functions, ODE solvers, time step sizes, and cellular events in CBMs. CBMOS [20] utilizes fast GPU vector operations provided by CuPy for efficient calculations and achieves a simulation speed 30X faster than their CPU version. Their emphasis, however, is on a better user interface for fast prototyping of new models. Its ability to handle large systems is still limited by the platform design, e.g., the force calculation time complexity is $O(N^2)$, and the GPU memory consumption can exceed 16 GB (e.g., NVIDIA Tesla T4) for $>10^4$ cells. GPU BioDynaMo [21] upgrades BioDynaMo [22] by enabling GPU co-processing. With a GTX 1080Ti GPU, GPU BioDynaMo can be 130X faster than the single thread CPU-only version BioDynaMo. However, GPU BioDynaMo does not solve PDEs on GPU, and its uniform grid method for force calculation still needs CPU for linked list maintenance, which significantly limits its real-world performance.

Therefore, although acceleration of cell simulation using GPU has been demonstrated, the potential has not been fully realized. Specifically, the cell-cell interaction has not been fully

parallelized, and the slow data transfer between CPU and GPU results in significant overhead. In this study, we develop a new open-source fast and memory-efficient fully GPU-based hybrid simulation software, GPU cell (Gell), to overcome these bottlenecks for large-scale hybrid cell simulation.

## Methods

### Cell model

**Cell cycle and death.** We model five cell phases in Gell. The premitotic, postmitotic, and quiescent phases are for living cells, and the necrotic and apoptotic phases are for cell death. The phase transition from premitotic to postmitotic and from postmitotic to quiescent are deterministic with a fixed gap time, respectively. Meanwhile, the phase transition from the quiescent phase to the premitotic phase and any phase transition from the living cell phase to the dead cell phase are all stochastic with a certain transition rate. The respective transition rates are listed in Table 1.

The probability for any stochastic transition α with transition rate ra to take place in a short time interval Δt is given by:

$$Prob_{\alpha} = r_a \Delta t. \tag{1}$$

During the quiescent phase, cells maintain their standard volume and remain to be stochastically activated for division preparation at the rate $r_{pro}(P_{oxy})$:

$$r_{pro}(P_{oxy}) = \begin{cases} r_{pro\_max} & P_{oxy} \geq Sa_{pro}, \\ r_{pro\_max} \dfrac{P_{oxy} - Th_{pro}}{Sa_{pro} - Th_{pro}} & Th_{pro} < P_{oxy} < Sa_{pro} \\ 0 & P_{oxy} \leq Th_{pro}. \end{cases} \tag{2}$$

The proliferation rate increases linearly with local oxygen concentration in the given range $Th_{pro}<P_{oxy}<Sa_{pro}$, where $Th_{pro}$ is the minimum oxygen partial pressure required for proliferation, $Sa_{pro}$ is the saturation oxygen partial pressure when the transition rate for proliferation reaches the maximum value $r_{pro\_max}$.

Once activated for proliferation, cells enter the premitotic phase and prepare for division. Premitotic cells gradually gain mass/volume and then divide at the end of this phase and enter the postmitotic phase. The division process is mechanically modeled to finish in an instant, and the two daughter cells equally inherit half of the parent cell volume and be placed around the parent cell center with the displacement $x_{disp}$ [15].

$$x_{disp} = \pm \left( R - \frac{R}{\sqrt[3]{2}} \right) \theta. \tag{3}$$

**Table 1. Phase transition in Gell.**

| Transition Rate | Premitotic Phase | Postmitotic Phase | Quiescent Phase | Apoptotic Phase | Necrotic Phase |
|---|---|---|---|---|---|
| Premitotic Phase | | $1/T_{prem}$ | 0 | 0 | 0 |
| Postmitotic Phase | 0 | | $1/T_{postm}$ | 0 | 0 |
| Quiescent Phase | $r_{pro}(p_{oxy})$ | 0 | | $r_{apop}$ | $r_{nec}(p_{oxy})$ |

The phase transition rate in the cell cycle model from the left column phase to the top row phase. The transition rate for deterministic phase duration is noted as one over the fixed phase duration.

Where R is the cell radius of the parent cell, and θ is a three-dimensional random unit vector. The postmitotic phase accounts for the duration required for daughter cells to reach mechanical equilibrium and grow to be proliferation ready.

Cell death is activated stochastically for all living cells. The transition rate to enter the apoptotic phase is a constant for all cells, $r_{apop}$, while the transition rate for necrotic death depends on the local oxygen partial pressure $P_{oxy}$:

$$r_{nec}(P_{oxy}) = \begin{cases} 0 & P_{oxy} \geq Th_{nec}, \\ r_{nec\_max} \dfrac{Th_{nec} - P_{oxy}}{Th_{nec} - Sa_{nec}} & Sa_{nec} < P_{oxy} < Th_{nec}, \\ r_{nec\_max} & P_{oxy} \leq Sa_{nec}. \end{cases} \tag{4}$$

Necrosis happens only when local oxygen partial pressure is lower than the necrosis threshold $Th_{nec}$. The transition rate increases linearly as the oxygen concentration decreases till the maximum necrosis rate $r_{nec\_max}$ is reached when oxygen partial pressure equals the necrosis saturation threshold $Sa_{nec}$.

All cell components start to shrink for apoptotic death upon entering the phase. While for the necrotic phase, early necrotic cells first absorb fluid and swell in the oncosis process, and then enter the late necrosis process after the membrane ruptures and start to lose fluid components. Modeling of two-stage necrotic death can be critical when studying the microstructures in the hanging drop spheroid necrotic center [15]. Phase transition-related parameters can be found in Table 2, where the listed values are all adopted from the "Ki67 Advanced" model from PhysiCell [15].

Following PhysiCell [15], we divide the total cell volume V into the fluid volume $V_F$ and the solid biomass volume $V_S$. The solid biomass volume is further divided into total nuclear volume $V_{NS}$ and cytoplasmatic volume $V_{CS}$. Different cell components have different rates of volume gain and loss in different phases. All the volume changes of different cell components are modeled using ordinary differential equations (ODEs):

$$\frac{dV_i(t)}{dt} = r_{p,i}(V_i(t) - V_i^p). \tag{5}$$

Where $V_i$ is the volume of component i, $r_{p,i}$ is the volume change rate of component i in phase p, and $V_i^p$ is the desired volume of component i in phase p. Specially, the desired fluid volume of a cell is a function of the current total cell volume V. Related parameters can be found in Table 3, adopted from MCF-10A human breast cancer cell line [15].

**Table 2. Phase transition-related parameters.**

| Parameter | Biophysical meaning | Reference Value |
|---|---|---|
| $T_{prem}$ | Duration of premitotic phase | 13 *hour* |
| $T_{postm}$ | Duration of postmitotic phase | 2.5 *hour* |
| $r_{apop}$ | Apoptosis rate | 0.0060 *hour*$^{-1}$ |
| $r_{pro\_max}$ | Max proliferation rate | 0.1176 *hour*$^{-1}$ |
| $Sa_{pro}$ | The oxygen level when the proliferation rate reaches maximum | 10 *mmHg* |
| $Th_{pro}$ | The oxygen level when the proliferation rate drops to zero | 5 *mmHg* |
| $r_{nec\_max}$ | Max necrosis rate | 0.1667 *hour*$^{-1}$ |
| $Sa_{nec}$ | The oxygen level when the necrosis rate reaches maximum | 2.5 *mmHg* |
| $Th_{nec}$ | The oxygen level when the necrosis rate drops to zero | 5 *mmHg* |

**Table 3. Cell volume growth related parameters.**

|  | Premitotic Phase | Postmitotic Phase/ Quiescent Phase | Apoptotic Phase | Early Necrotic Phase | Late Necrotic Phase |
|---|---|---|---|---|---|
| $V_F$ | $0.7502V$ | $0.7502V$ | $0\ um^3$ | $V$ | $0\ um^3$ |
| $r_F$ | $3\ hour^{-1}$ | $3\ hour^{-1}$ | $3\ hour^{-1}$ | $0.67\ hour^{-1}$ | $0.05\ hour^{-1}$ |
| $V_{NS}$ | $270\ um^3$ | $135\ um^3$ | $0\ um^3$ | $0\ um^3$ | |
| $r_{NS}$ | $0.33\ hour^{-1}$ | $0.33\ hour^{-1}$ | $0.35\ hour^{-1}$ | $0.013\ hour^{-1}$ | |
| $V_{CS}$ | $976\ um^3$ | $488\ um^3$ | $0\ um^3$ | $0\ um^3$ | |
| $r_{CS}$ | $0.27\ hour^{-1}$ | $0.33\ hour^{-1}$ | $1\ hour^{-1}$ | $0.0032\ hour^{-1}$ | |

**Cell mechanics.** Cells in Gell are mechanically modeled as elastic balls with varied volumes and center positions. Cells adhere to each other while attached and push against each other upon compression. The motion of cell i at position $x_i(t)$, with velocity $v_i(t)$, and with a set $N_i(t)$ of nearby cells can be modeled as [15]:

$$m_i \dot{v}_i = \sum_{j \in N(i)} (F_{cca}^{ij} + F_{ccr}^{ij}) + F_{mot}^{i} - \eta_i v_i. \tag{6}$$

Where $\dot{v}_i$ is the acceleration, $F_{cca}^{ij}$ denotes the adhesive force from cell j to cell i, and $F_{ccr}^{ij}$ represents the repulsive force from cell j to cell i. $F_{mot}^{i}$ accounts for the force related to cell migration. $\eta_i v_i$ represents the resistance contributed by the local microenvironment, such as fluid resistance and cell-matrix adhesion forces. $\eta_i$ is a fluid-drag-coefficient parameter, and $v_i$ is the cell velocity.

The force equilibrates at relatively short time scales relative to the time scale of cell volume change and multicellular patterning. Therefore, we can safely apply the zero acceleration inertialess condition to Eq (6) and explicitly solve $v_i$ by:

$$v_i = \frac{1}{\eta_i} \left( \sum_{j \in N(i)} (F_{cca}^{ij} + F_{ccr}^{ij}) + F_{mot}^{i} \right). \tag{7}$$

The adhesive force and repulsive force experienced by cell i are modeled as:

$$F_{cca}^{ij} = \begin{cases} C_{cca} \left( 1 - \frac{|r_{ij}|}{R_A} \right)^2 \frac{r_{ij}}{|r_{ij}|} & if\ |r| \leq R_A \\ 0 & otherwise. \end{cases} \tag{8}$$

$$F_{ccr}^{ij} = \begin{cases} -C_{ccr} \left( 1 - \frac{|r_{ij}|}{R_R} \right)^2 \frac{r_{ij}}{|r_{ij}|} & if\ |r| \leq R_R \\ 0 & otherwise. \end{cases} \tag{9}$$

$R_R$ is the maximum repulsive interaction distance that equals the sum of the radius of cell i and j. $R_A$ is the maximum adhesive interaction distance, which is slightly larger than $R_R$ due to the deformability of the two cells. $C_{cca}$ is the cell-cell adhesion parameter, and $C_{ccr}$ is the cell-cell repulsion parameter. $r_{ij}$ is a vector pointing from the center of cell i to the center of cell j.

Once the sum of the experienced force is calculated for all cells, the velocity of any cell can be directly calculated. Cell position is then updated using the second-order Adam-Bashforth method:

$$x_i(t + \Delta t) = x_i(t) + \frac{\Delta t}{2} (3 \cdot v_i(t) - v_i(t - \Delta t)). \tag{10}$$

## Extracellular microenvironment

The tumor is surrounded by a complex ecosystem named tumor microenvironment (TME), composed of tumor cells, stromal cells, and other extracellular physical and chemical factors. The mutual and dynamic crosstalk between the tumor and tumor microenvironment, together with the genetic/epigenetic change in tumor cells, are two factors that influence the formation and progression of the tumor [23]. In our model, cells can absorb environmental nutrients and release biochemical factors into the extracellular fluid. In addition, critical environmental factors can also regulate cell behaviors. To simulate the spatio-temporal variation of environmental factors during tumor development, we consider a continuous extracellular fluid space and use PDEs to describe the secretion, diffusion, uptake, and decay of diffusive substances such as oxygen and vascular endothelial growth factor. The continuum environment and the discrete cells are explicitly linked. Cell phases, sizes, and positions are treated as static while updating the continuous molecular space and vice versa. The equation for any diffusive substance in the extracellular fluid domain $\Omega$ can be written as [24]

$$\frac{\partial \rho}{\partial t} = \nabla \cdot (D\nabla\rho) - \lambda\rho + S(\rho^* - \rho) - U\rho. \tag{11}$$

Depending on the problem, the domain boundary $\partial\Omega$ can be either Dirichlet or Neumann type. $\rho$ is the substance concentration, $\rho^*$ is the saturation concentration, $D$ is the diffusion coefficient, $\lambda$ is the decay rate, $S$ is the supply rate, $U$ is the uptake rate.

In the provided code, the oxygen concentration is the only considered environmental factor and discrete cells are the only contributor of oxygen consumption. Cells absorb oxygen from the extracellular fluid and regulate their behavior according to local oxygen concentration. The oxygen consumption of each cell is modeled as proportional to both the oxygen concentration and the cell volume. With a cartesian grid, for each isotropic voxel i, the total uptake rate of oxygen equals the sum of the local cell consumption:

$$U_i = \sum_{j \ in \ i} \frac{V_j}{V_{voxel}} U_o. \tag{12}$$

The uptake rate here represents the oxygen concentration decrease rate as a proportion of current concentration, $U_o$ is the default oxygen consumption rate of living cells that equals 10 per min. $V_{voxel}$ is the volume of the given voxel, $j$ is the index of cells inside the voxel, and $V_j$ is the corresponding cell volume.

In the simulation, each voxel's total oxygen consumption rate is first calculated according to the position, size, and phase of all the cells. Then the molecular space is updated using the static consumption rate map. With the calculated molecular concentration, cells in the discrete model could read the local oxygen concentration and carry on their stochastic phase transitions according to these values.

For the numerical processing of the PDE, following BioFVM [24], a first-order splitting method is first applied to split the righthand side into simpler operators: a supply and uptake operator and a diffusion-decay operator.

$$\begin{cases} \dfrac{\sigma - \rho^n}{\Delta t} = \nabla \cdot (D\nabla\sigma) - \lambda\sigma, \\ \dfrac{\rho^{n+1} - \sigma}{\Delta t} = S(\rho^{n+1} - \sigma) - U\sigma. \end{cases} \tag{13}$$

The supply and uptake operators are handled analytically. The three-dimensional diffusion-decay operator is further split into a series of related one-dimensional PDEs using the

locally-one dimensional (LOD) method.

$$\begin{cases} \dfrac{\eta - \rho^n}{\Delta t} = \partial_x(D\partial_x\eta) - \dfrac{1}{3}\lambda\eta, \\ \dfrac{\eta^* - \eta}{\Delta t} = \partial_y\left(D\partial_y\eta^*\right) - \dfrac{1}{3}\lambda\eta^*, \\ \dfrac{\sigma - \eta^*}{\Delta t} = \partial_z(D\partial_z\sigma) - \dfrac{1}{3}\lambda\sigma. \end{cases} \quad (14)$$

Discretized using the finite volume method, we obtain the updated concentration of each strip of voxels for each direction by solving Eq (15) using the Thomas algorithm [25].

$$\begin{cases} \left(1 + \dfrac{1}{3}\Delta t\lambda + \dfrac{\Delta t}{\Delta x^2}D\right)^{\circ}\eta_{(0,y_f,z_f)} - \dfrac{\Delta t}{\Delta x^2}D^{\circ}\eta_{(1,y_f,z_f)} = \rho^n_{(0,y_f,z_f)} \\ -\dfrac{\Delta t}{\Delta x^2}D^{\circ}\eta_{(n_x-1,y_f,z_f)} + \left(1 + \dfrac{1}{3}\Delta t\lambda + 2\dfrac{\Delta t}{\Delta x^2}D\right)^{\circ}\eta_{(n_x,y_f,z_f)} - \dfrac{\Delta t}{\Delta x^2}D^{\circ}\eta_{(n_x+1,y_f,z_f)} = \rho^n_{(n_x,y_f,z_f)} \\ -\dfrac{\Delta t}{\Delta x^2}D^{\circ}\eta_{(N_x-2,y_f,z_f)} + \left(1 + \dfrac{1}{3}\Delta t\lambda + \dfrac{\Delta t}{\Delta x^2}D\right)^{\circ}\eta_{(N_x-1,y_f,z_f)} = \rho^n_{(N_x-1,y_f,z_f)}. \end{cases} \quad (15)$$

## Implementation details

Gell is developed using C++ and CUDA (v11.2), with the program's design schematic illustrated in Fig 1. There are two major types of data in the simulation, the pre-allocated array of structure for cell data management and the isotopically discretized cartesian grid for spatial domain-related data, which includes information related to the tumor microenvironment (TME) and our Voxel Sorting (VS) method. Once initialized on the CPU, all the cellular and microenvironmental data are transferred to GPU memory. From this point, all the following computation steps are exclusively handled by GPU to eliminate costly back-and-forth data transfer between GPU and CPU.

In the provided diagram, white cylinders symbolize the simulation data, and colored boxes positioned atop this data represent associated operations. The abbreviations TME and VS are used to denote the Tumor Microenvironment and Voxel Sorting, respectively. The

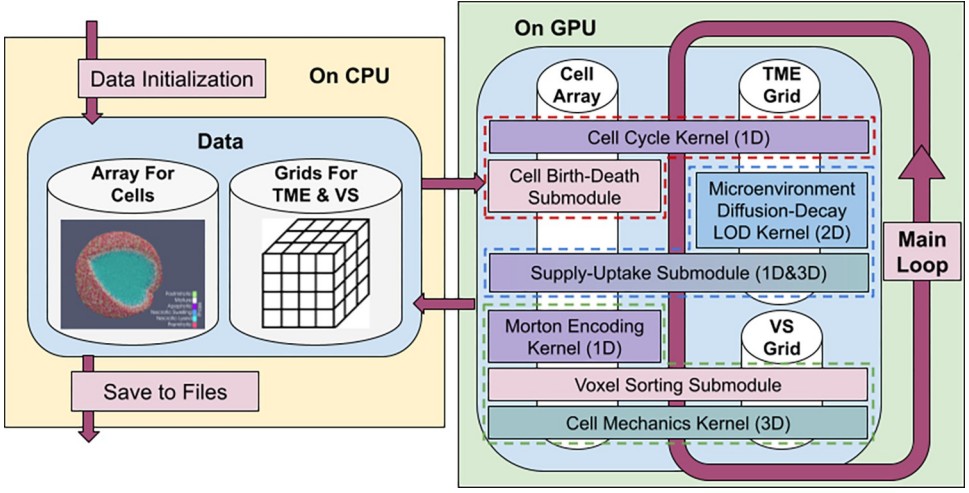

**Fig 1. Diagram for the program design.**

Microenvironment Module, Cell Cycle Module, and Cell Mechanics Module are distinctly demarcated with dashed lines in blue, red, and green, respectively.

The main simulation loop of Gell consists of three critical computational modules, as delineated in dashed lines in Fig 1: the Microenvironment Module, the Cell Cycle Module, and the Cell Mechanics Module. Each module is responsible for one of the key tasks in the hybrid cell-based simulation and is uniquely designed to enable efficient parallelization on the GPU. Further details on this these modules will be explored in the following sections.

**Microenvironment module.** The Microenvironment Module is tasked with managing the diffusive transportation and cellular secretion/uptake of environmental factors, such as oxygen. This module is divided into two sections: a Supply-Uptake Submodule and a Diffusion-Decay LOD Kernel.

The Supply-Uptake Submodule deals with the supply and uptake of microenvironmental factors. This process involves two CUDA kernels: a 1D kernel parallelized over the cells to calculate each voxel's total supply and uptake, followed by a 3D kernel parallelized over voxels to adjust the factor concentration after supply and uptake.

The Diffusion-Decay LOD Kernel addresses the diffusion and decay of environmental factors using the LOD algorithm. During each diffusion-reaction step, the LOD solver transforms the concentration update problem along the three axes into $3*N^2$ tridiagonal linear systems, each with N unknowns. Each parallel thread solves one of these tridiagonal linear systems using the Thomas algorithm. The overall computational complexity of each update step is O($N_{voxel}$).

**Cell cycle update.** The Cell Cycle Module is responsible for managing cell proliferation, growth, and death. It is divided into two components, the cell cycle kernel, and the cell birth-death submodule.

The cell growth and phase transition only involve the current cell status as well as the local environmental factors. These processes are highly parallelizable and are directly distributed across different threads and computed by GPU in the Cell Cycle Kernel. The Cell Birth-Death Submodule manages cell proliferation and death, focusing on the memory management of structured data. By leveraging this submodule, fast simulations can be conducted with only minimal additional computation and memory cost for thread competition avoidance. During cell proliferation, the total cell count is atomically updated to enable the addition of new daughter cells to the cell array. For cell death, cells earmarked for removal are initially labeled in their respective phase update thread and then collectively deleted from the cell array in a subsequent kernel for efficient cell array maintenance. The overall computational complexity of the above-mentioned processes is approximately O(N).

**Cell mechanics update.** N-body interaction simulations can be extremely expensive due to their O($N_{cell}^2$) computational complexity. For a large multicellular system with millions of cells, it is computationally impractical just to loop over all the cell pairs, even with GPU [18, 20]. Fortunately, the cell-cell mechanical interactions are short-range interactions, making it reasonable to calculate only the forces between neighboring cells, thus reducing the computational complexity to O($N_{cell}$). PhysiCell utilizes the cell-cell interaction data structure (IDS) method [15]. A large number of lists are created and maintained to record the indices of cells inside each voxel. The force calculation for each cell only has to loop over the cells inside its nearest 27 voxels, according to the lists. GPU BioDynaMo, with its uniform grid method [21], improves memory usage efficiency by replacing the cell index list with the linked list. However, maintaining such cell lists or linked lists is not GPU-friendly. GPU does not allow us to dynamically allocate memory in the thread, making it challenging to create lists with dynamic lengths for all mechanical voxels in the IDS method. Suppose GPU memory for cell lists is all pre-allocated according to the maximum cell density. In that case, memory usage can be highly

inefficient due to the high cell density necrotic region. The challenge lies in the maintenance of the linked list for the uniform grid method on GPU. Thread locks are required to update the linked list correctly, but the wrapping mechanism of CUDA could easily create deadlocks during such processes and pause the program indefinitely.

To realize an efficient cell-cell mechanics computation on GPU, we have developed our Voxel-Sorting Method. Cells are stored in array-of-structures, and a Morton code is generated for each cell according to the i, j, k index of its containing mechanical voxel as a key for sorting. Then a fast GPU-based radix sort algorithm [26] of complexity $O(N_{cell})$ is used to rearrange the cell array according to the ascending Morton code value order. After sorting, cells in the same voxel are stored contiguously in the GPU memory, and cells in adjacent voxels are stored relatively close. Such a contiguous memory layout increases the memory fetch efficiency, especially when groups of cells in the same voxel have to be accessed together by the same thread. The array index range for cells inside each voxel is easily determined with $O(N_{voxel})$. Fig 2 illustrates the voxel sorting method in a 2D scene.

This rapid cell-cell interaction simulation method described above is implemented in the three-step Cell Mechanics Module. The first Morton Encoding kernel determines the order index of each cell. Then the second Voxel Sorting Submodule sorts the cells accordingly and determines the index range of cells within each voxel. In the last step, the aggregated force and cell movements of each cell are calculated by the Cell Mechanics Submodule.

**Time scale considerations.** Different biological processes evolve at different time scales: the temporal scale of cell colony biology and mechanics is on the order of minutes, while the equilibrium of transport diffusion is achieved in seconds [15]. Because the extracellular environment updates substantially faster than cellular evolution, it is computationally inefficient to

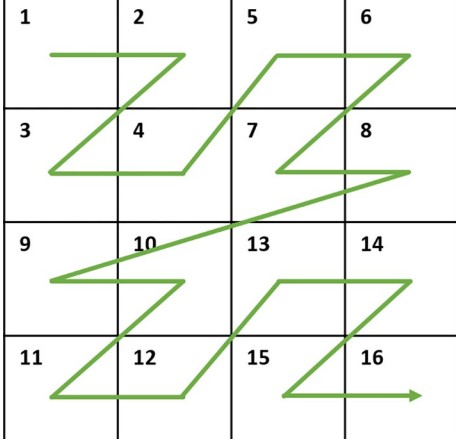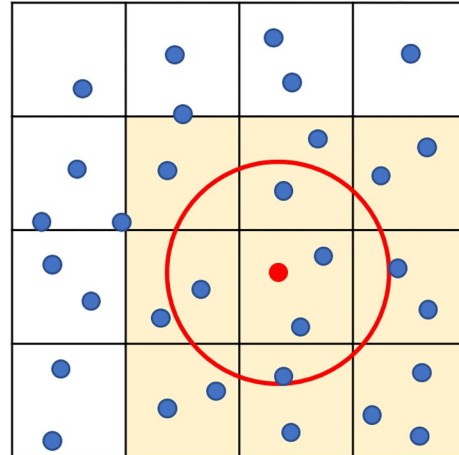

**Fig 2. 2D illustrations of voxel sorting method.** Left: The Morton code maps 2D position to 1D number while preserving the locality of the data points. The Morton code is generated by interleaving the binary x and y pixel index values. Adapted from [27]. Right: The force calculation example of a single cell(red). The red circle represents the maximum range of cell-cell mechanical interaction whose radius is shorter than the side length of voxels. The Force module only needs to loop over the cells in yellow voxels to calculate the aggregated force, and the indexes of cells inside these voxels can be easily figured out after the sorting. As long as the voxel side length is longer than the maximum cell-cell interaction distance, each cell's aggregated cell-cell mechanical interaction can be calculated efficiently by looping only over cells inside its nearest 27 voxels. Our voxel sorting method achieved force calculation time complexity of $O(N_{cell})$ and high GPU memory utilization efficiency. An additional advantage of our voxel sorting method is memory access efficiency. With the cells in the same voxel stored next to each other in the GPU memory after voxel sorting, the following 27 times of memory fetch of these cell data can become much more efficient than fully random memory access. After the force calculation, the cell position update can be directly parallelized using the aggregated force information.

synchronize the cell simulator and extracellular environment simulator updates. Instead, we first fixed cellular properties while solving the PDEs for the extracellular environment at a higher frequency and then evolved cell phenotype and mechanical interaction at a lower frequency to reduce the simulation cost.

## Results

All calculations are tested on a personal computer (with Intel® Core™ i7-7700K CPU, 64GB memory, and an NVIDIA® GeForce® RTX 2080Ti graphics card) on 64-bit Windows 10. Tests are conducted to determine the optimal block size settings for GPU computation during the program development, which have been fixed for later simulations. Specifically, the kernels for cell cycle update have a 1D block width of 64, while the 3D kernels for oxygen consumption employ a block width of 4. As for the 2D kernels used for LOD calculation, we have adopted a small 2D block width of 4 due to the storage requirements of the Thomas algorithm on GPU. This algorithm necessitates the storage of a temporary array for each thread, and a small block size allows us to store all data on the registers.

### Hanging drop spheroid

As an *in vitro* 3D multicellular model, multicellular tumor spheroids (MCTS) possess many *in vivo* tumor features, including cell-cell interaction, hypoxia, treatment response, and production of extracellular matrix [28, 29]. Tumor spheroids are widely used for various tumor growth dynamics and treatment response studies. Many modeling works have been published to mathematically bridge the observed spheroid experiment results to mechanistic understandings. Joshua A. Bull et al. [30] developed an off-lattice hybrid spheroid growth model to explore the growth dynamics of tumor spheroids and reproduced the migration and internalization of tumor cells observed in spheroid experiments [31]. Kevin O. Hicks et al. [32] developed an on-lattice hybrid spheroid model using experimentally determined parameters and accurately simulated the cell killing after radiation and hypoxia-activated prodrug interventions.

To compare the computational performance of our simulation framework with existing simulation software, we simulated a benchmark problem of hanging drop spheroid growth [15]. In the hanging drop spheroid (HDS) benchmark, a suspended multicellular aggregate is cultured in the middle of a growth medium with oxygen supplied through diffusion from the domain boundary. All the Gell simulations and PhysiCell simulations share identical simulation settings. The simulation started with 2347 cells and evolved to nearly one million cells after 450 hours of cultivation. The simulation domain contains one million isotropic voxels with a side length of 25 μm. Cell mechanics and phase update is calculated every 0.1 minutes, and the diffusion-reaction of oxygen in the extracellular fluid is solved every 0.01 minutes. At the end of the simulation, Gell and PhysiCell produce nearly identical results, predicting a spheroid radius of 1.87 mm and evolving visually identical spheroid structures, including the crack patterns in the necrotic cores. These findings suggest that the computation of Gell is accurate, and the use of single precision arithmetic has a negligible impact on the simulation result. The evolution of cell number and spheroid radius over time is shown in Fig 3. The result shows no necrosis development until the spheroid reaches a certain radius when oxygen diffusion from the outer rim becomes insufficient to support the inner cells. The radius growth curve exhibits a linear shape due to the approximately constant viable rim thickness, which agrees with the simulation result and theoretical prediction of PhysiCell [15].

The rendered images of the simulation results are shown in Fig 4. There is a clear, viable rim of actively proliferating cells at the outer shell of the spheroid and a necrotic core in the

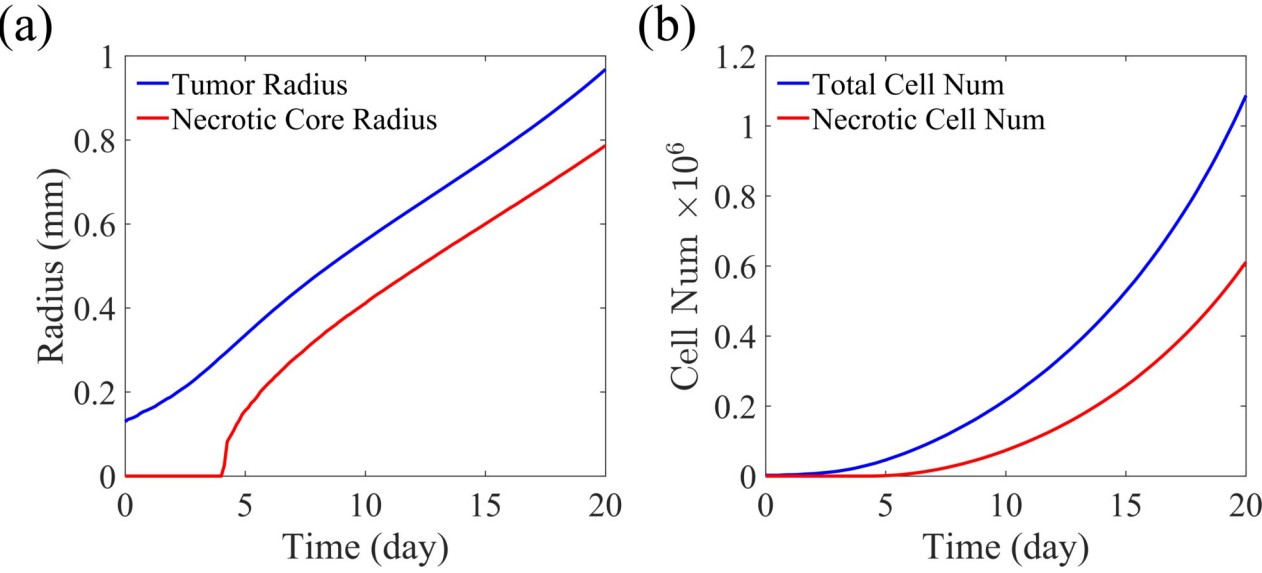

**Fig 3. Simulation result analysis of Gell.** (a) Whole tumor radius and necrotic core radius change over the HDS growth. (b) Number change of total tumor cells and necrotic tumor cells over the HDS growth.

center (Fig 4A). Additionally, the subtle cell-cell mechanical adhesion and crack-like micro-structures successfully emerged in the necrotic core entry. The simulation results of Gell agree with PhysiCell and *in vitro* experiments [15].

Gell completed the entire simulation process in 47 minutes using the personal computer. As a comparison, the state-of-the-art CPU-based paralleled simulator PhysiCell used 119 hours for the same simulation on the same personal computer. In other words, Gell is 150X faster for the cell simulation problem of this scale.

Benefiting from the accelerated computation, we were able to explore the parameter space without agonizing pain. We surprisingly found that the crack pattern of the necrotic core has little to do with the two-stage necrosis process of tumor cells. The central slice of the spheroid of non-swelling tumor cells shows almost identical microstructure patterns (Fig 5B). However, this system ends up with a slightly smaller size, fewer cells, and a higher overall necrotic debris

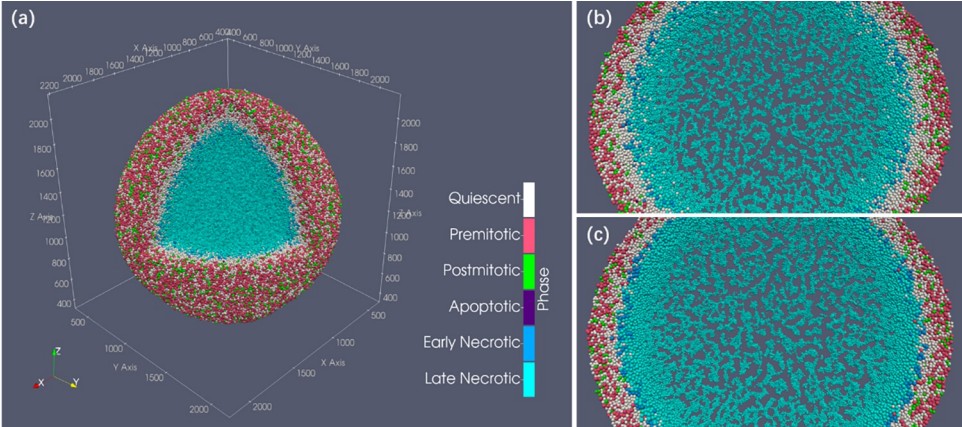

**Fig 4. HDS simulation result after 450 hours.** (a) Cell cluster generated by Gell. (b) 60 μm thick central slice of the HDS simulation result shows the microstructure of the necrotic core of Gell simulation. (c) central slice of PhysiCell showing identical microstructure. Both spheroids have a radius of 1.87 mm.

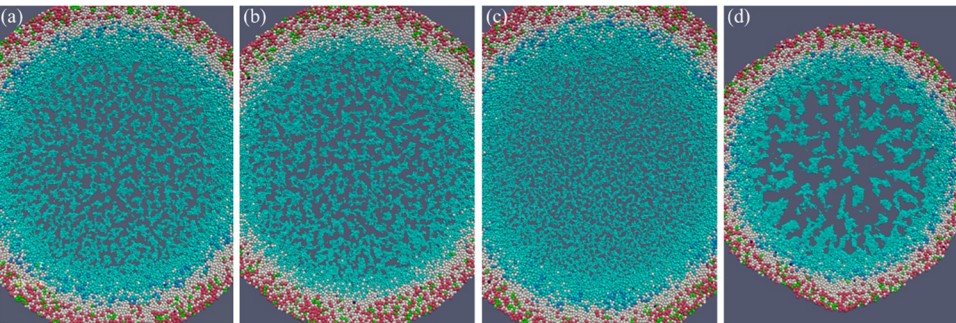

**Fig 5. HDS simulation with altered phenotype.** The 60-um thick central slices of simulated spheroids with various cellular mechanical properties. All the spheroids start with a small cluster of 2347 randomly placed cells, and the cultivation duration is 450 hours. (a) The reference spheroid ends up with 0.9 million cells and a diameter of 1.87 mm. (b) Spheroid of tumor cells with no swelling during early necrosis, with 0.9 million cells and a diameter of 1.8 mm. (c) Spheroid of tumor cells with the cell-cell adhesion suppressed, with 1.0 million cells and a diameter of 1.97 mm. (d) Spheroid of tumor cells with the cell-cell adhesion enhanced, with 0.66 million cells and a diameter of 1.53 mm.

density. This suggests that the swelling process of tumor cells could facilitate spheroid growth by pushing the viable cells toward the more oxygenated outer regions. Fig 5C shows the spheroid of tumor cells with the cell-cell adhesion suppressed. The cell-cell adhesion factor $C_{cca}$ is decreased to one-fourth of the reference value. The weak adhesion discourages gathering necrotic cells, leading to a more scattered distribution of smaller necrotic cell clusters with more minor interleaving cracks. Fig 5D is a spheroid with the cell-cell adhesion enhanced by quadrupling Ccca. The strong adhesive force helps form the massive necrotic debris clusters and promotes a significantly higher cell density that intensifies the oxygen competition between tumor cells and ultimately hinders tumor growth. Such pattern differences in necrotic core microstructures that emerged in the simulation are also observed in *in vitro* experiments [33], as shown in Fig 6. Two spheroids of the same melanoma cell line (A2508) form clustered (Fig 6A) and scattered (Fig 6A) necrotic cores, respectively. The exact differences in cell treatment and mechanisms of pattern formation are not described in the original literature. However, our simulations hint that cell-cell adhesion could be an important factor that dramatically affects the spheroid morphology, especially the necrotic core microstructure.

Simulations have the potential to depict a causal and mechanistic path from certain microscopic cell properties to the development of qualitative macroscopic morphology and provide

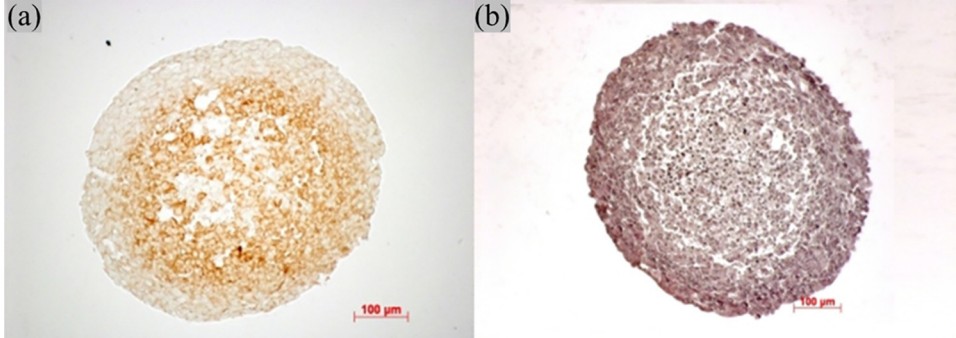

**Fig 6. Melanoma cell line spheroids.** Two spheroids of the same melanoma cell line (A2508) show distinct pattern differences in necrotic core microstructures due to differences in cell treatment. Images are adapted from [33], and treatment details are not mentioned in the original literature. (a) A pimonidazole stained spheroid. (b) A hematoxylin and eosin stained spheroid. Adapted from [33] with permission.

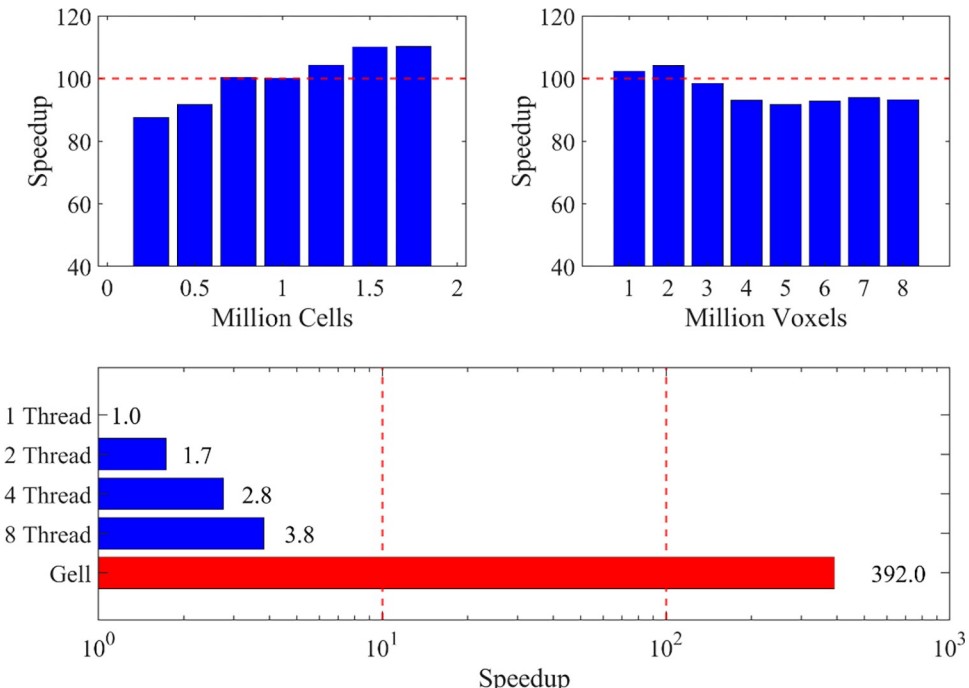

**Fig 7. Gell simulation speedup with respect to PhysiCell.** Gell simulation speedup with respect to PhysiCell with varying cell numbers (a), domain voxel numbers (b), and PhysiCell CPU thread numbers (c with logarithmic x scale).

insights into real-world phenomena. However, exploring the parameter space and improving the models iteratively can be very time-consuming. Gell could help these studies by dramatically increasing the computational speed.

Besides the baseline HDS simulation task, further comparisons of simulation performance with varying initialized cell numbers (Fig 7A) and domain sizes (Fig 7B) suggest that Gell is consistently around two orders of magnitude faster than multi-thread PhysiCell on the personal computer. For serial PhysiCell using only one thread, Gell can be almost 400x faster (Fig 7C). The default simulation setting for the performance comparison contains one million 25×25×25 μm isotropic voxels and one million living cells. Cells are randomly initialized in a sphere with a specific cell density. The acceleration ratios of these tests differ from the whole HDS simulation because the HDS simulation has a more heterogeneous cell distribution and is closer to the equilibrium states after prolonged mechanical interactions.

Additionally, Gell can complete the simulation with its maximum memory footprint of only one-tenth of PhysiCell without sacrificing accuracy and system complexity (Table 4).

**Table 4. Memory footprint comparison.**

|  | PhysiCell | Gell |
|---|---|---|
| Memory Footprint of HDS Simulation (MB) | 6360 | 500 |
| Memory Footprint per Additional Million Cells (MB) | 5384.2 | 118.0 |
| Memory Footprint per Additional Million Voxels (MB) | 742.8 | 15.38 |

Memory footprint comparison of Gell and PhysiCell per additional million cells and per additional million domain voxels in the HDS simulation task. Our model is able to simulate the problem with a much lower memory occupation.

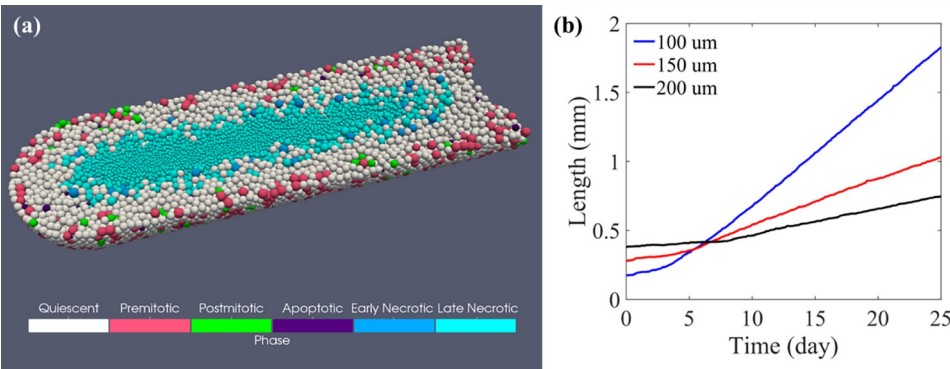

**Fig 8. Simulation results of DCIS development.** (a) Ductal carcinoma in situ simulation with duct radius of 150 μm. (b) Linear DCIS growth under various duct radius conditions.

## Ductal carcinoma in situ

Ductal carcinoma in situ (DCIS) is non-invasive breast cancer that grows within the lumens of the mammary duct [34]. DCIS itself is not hard to treat, but as a precursor to invasive ductal carcinoma with a high incidence rate (26.6 per 100000 women [35]), its development and progression raise the interest of many modelers [36–40]. Following the work of Paul Macklin (30), we created a hybrid DCIS simulation example to illustrate Gell with a more complex task further. A cluster of tumor cells is placed at the dead-end of a fixed single-opening duct. The movement of tumor cells is confined within the tube lumen, and the oxygen is supplied through diffusion across the duct wall into the lumen. Breast ducts have a typical radius ranging from 100 μm to 200 μm [38]; therefore, we simulated three growth scenarios of ductal carcinomas in situ with the respective duct radius of 100 μm, 150 μm, and 200 μm.

With the same cell model as in the HDS simulation and a Dirichlet boundary condition of 7.2 mmHg oxygen concentration applied to the duct surface, the average rate of DCIS advance for ducts of radius 100, 150, and 200 μm is 77.0, 32.7, 19.0 μm/day, respectively. Our simulation results in Fig 8 show a linear growth speed of DCIS, and the tumor advance rate has an inverse relationship with the duct radius, which agrees with the clinical observations and other computational studies in 2D [36] or 3D [41].

## Performance testing

**Individual module time cost.** We first evaluated our simulator with a randomly initiated spherical cell cluster with one million cells for performance assessment. The simulation domain contains one million 25 μm-long isotropic voxels. The time cost of each module is listed in Table 5. With the entire calculation paralleled on GPU, Gell maximizes computational efficiency in all the simulation modules. Meanwhile, unnecessary data transfer between CPU and GPU during the simulation is eliminated, ensuring a low delay between modules.

In the table, the cell-sorting-related process appears to be slow, but in practice, its impact on the overall computation is small. Firstly, the cell sorting module is less invoked than many other modules. The simulation faces a multiscale problem. The cell phase is updated every few simulation minutes, cell motion is updated every few seconds, and the diffusion-reaction of oxygen is updated more than once per second. In this case, the simulation is approximately equally dominated by the cell-motion-related modules and the diffusion-reaction model. Secondly, the sorting process accelerates the rest of the calculations. Take the oxygen consumption model as an example. This kernel is a part of the reaction-diffusion module that calculates

**Table 5. The execution time cost of each simulation module in Gell.**

| Module | Time cost per invocation |
|--------|--------------------------|
| Morton code calculation | 0.303 ms |
| Cell sorting | 6.041 ms |
| Force calculation | 3.613 ms |
| Cell movement | 0.360 ms |
| Reaction-diffusion update | 0.542 ms |
| Phase update with birth and death | 4.744 ms |
| Memory copy between CPU and GPU | 0.011 ns |
| Others | 0.387 ms |

Values are averaged over 1000 simulation steps with the time step for phase update, mechanics update, and diffusion-reaction update all set to be the same. Short time such as the time cost of memory copy during a simulation, is estimated by comparing a 1000-step simulation with another 2000-step simulation. The time cost for initialization and data saving is omitted.

each cell's oxygen consumption rate and adds the value to the total consumption rate of each corresponding voxel. This kernel works with both unsorted and sorted cell structures. Changing from unsorted random memory access to sorted contiguous memory access, this module's time cost per invocation is reduced from 0.345 ms to 0.170 ms by 51%. The force calculation module is expected to benefit most from the high data fetch efficiency of the voxel sorting method. Because force calculation does not work with unsorted data structures, the speed comparison cannot be performed. Nevertheless, the longer time spent on cell sorting benefits the overall computation and is a worthy investment.

## Performance scaling

Unaffordable memory occupation is another potential barrier for large-scale cell-based simulations. Memory efficiency is another design goal to make Gell a suitable software for large-scale simulations on widely available devices. The GPU memory footprint of the HDS simulation with one million voxels and one million cells is limited to 500 MB, and the memory occupation increases with the cell number and domain size is linear and slow, as shown in Table 5. This enables Gell to fit extremely large-scale problems into a modern personal computer easily.

We tested Gell's ability to handle ultra-large-scale problems with a hypothetical ultra-large spheroid model. In reality, the size of hanging droplets is diffusion constrained. Larger *in vivo* tumors inevitably involve angiogenesis and supporting tumor vasculature. However, angiogenesis and oxygen transportation simulations are beyond the scope of the current work. Instead, we simulate a series of hypothetical huge hanging droplet development problems in a huge domain with varying initial cell numbers, each for one hour. As shown in Fig 9, Gell has a linear computational cost scaling with the cell number. Simultaneously, the Gell GPU memory footprint peaked at 4392 MB, showing high efficiency in memory and computational source usage. This linear time complexity and low memory occupation demonstrate that Gell can handle potential ultra-large-scale problems.

Time and memory cost for one hour's mechano-biological process simulation with varying cell numbers. The domain contains 250×250×250 25 μm-long isotropic voxels.

## Discussion

Cell simulation is driven by the need to model larger, more complex digital tumors parallel to human tumors containing billions of cells. The modeling accuracy of the biology and

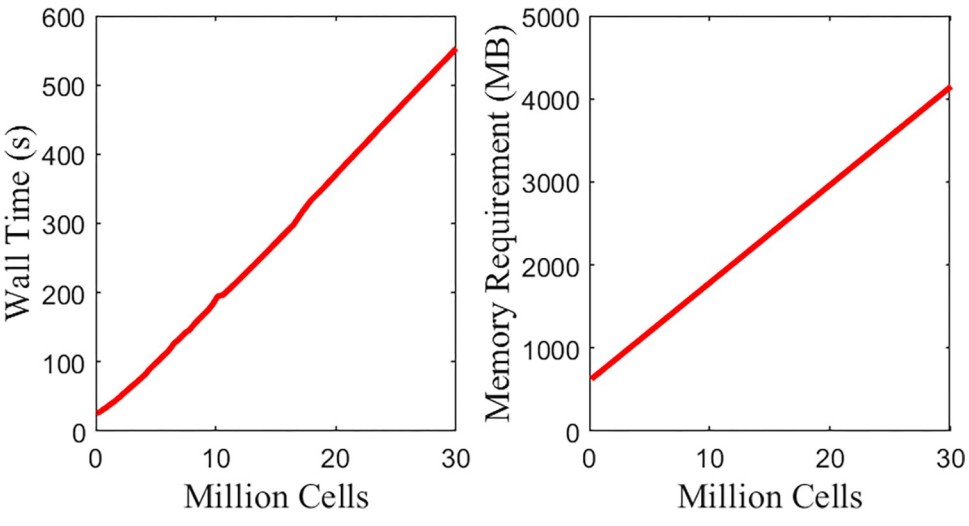

**Fig 9. Performance scaling of Gell.**

computational capacities has limited the simulation of such complex systems. Existing tools such as Biocellion [16] for such large-scale simulations are close-sourced and require expensive CPU clusters. Open source tools, including PhysiCell [15] and BioDynaMo [17] require high-end CPU clusters to perform computation in practical time. In theory, GPU is well suited to manage the large parallel components of cell simulation. However, due to the differences between GPU and CPU architecture, a direct translation of a CPU-based cell simulation code to a GPU-based one is not straightforward and can be inefficient. Specifically, global cell-cell interactions have a time complexity of $O(N^2)$. The nonlinear computational cost severely limits the number of cells that can be modeled and, subsequently, the size and complexity of the model. We need to maintain a neighboring cell list to overcome the challenge, which is difficult for GPU memory due to its dynamic nature. Besides this challenge, a incomplete translation of computation from CPU to GPU requiring frequent data transfer between them can also be rate limiting. As a result, the threshold of performing cell simulation to a size that is relevant to the small tissue scale remains out of reach for many biological researchers even with modern GPU architecture.

To overcome these challenges, we employed several novel computational techniques to fully leverage the GPU architecture, improve simulation performance, and minimize memory usage.

As the computational cost of cell-cell interaction rises quadratically with the cell number, we developed a GPU-friendly voxel sorting method that handles the short-range cell-cell interaction modeling and improves the simulation's memory access efficiency. We also implemented a fully GPU-based LOD solver for the spatiotemporal variation of diffusive substance distribution in extracellular water. As a result, we optimized the evolution algorithm to achieve linear computational complexity $O(N)$ while minimizing the memory footprint. In the numerical implementation, we fully exploited the parallel architecture of modern GPU and different types of GPU memory for high computational speed and low memory access overhead. The computation is nearly 100% on GPU, thus avoiding slow data transfer between CPU and GPU memories.

Our GPU implementation significantly outperformed CPU methods, leading to almost 400X speedup over the single-thread version of the well-established CPU simulator on a personal computer. The acceleration is the highest among existing GPU-powered simulators. The

acceleration, in combination with the low memory footprint, makes Gell readily available to biology researchers. The easy-to-access platform would facilitate the fast prototyping of hybrid models and hypothesis testing for large-scale problems.

As a future research direction, Gell can be scaled to multiple GPUs for larger problems on the order of $10^9$ cells. As previously alluded, simulation of tumor vasculature and other scaffolding cells such as the stromal and immune cells, would be necessary for a biologically relevant model. Besides our effort to incorporate these extremely complex biological processes for digital tumor twins, we support our peers to join the effort by providing and updating Gell as a user-friendly open-source tool. The source code can be found at https://github.com/PhantomOtter/Gell.

## Conclusion

Large-scale cell simulations are valuable for hypothesis generation and testing experimental parameters. However, the high computational cost of the CPU-based simulation has limited the simulation size and practicality. In the current study, we describe a novel GPU cell simulation platform Gell to fully leverage the highly parallel nature of GPU. For the first time, we demonstrated a GPU-friendly voxel sorting method that reduced the quadratic cell-cell interaction computational complexity to be linear. The full GPU implementation avoided unnecessary CPU-GPU data transfer overhead. As a result, Gell achieved a 400X acceleration and 1–2 order of magnitude reduction in the memory footprint compared to a state-of-the-art CPU cell simulation platform, PhysicCell, for the same hanging droplet tumor spheroid and ductal carcinoma in situ simulation tasks.

## Author Contributions

**Conceptualization:** Ke Sheng.

**Data curation:** Jiayi Du.

**Formal analysis:** Jiayi Du.

**Funding acquisition:** Ke Sheng.

**Methodology:** Jiayi Du, Yu Zhou, Lihua Jin, Ke Sheng.

**Software:** Jiayi Du.

**Supervision:** Ke Sheng.

**Visualization:** Jiayi Du, Yu Zhou.

**Writing – original draft:** Jiayi Du.

**Writing – review & editing:** Yu Zhou, Lihua Jin, Ke Sheng.

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
