## [Decision Letter · Decision Letter 0]

2 May 2023

PONE-D-23-02925Gell: A GPU-powered 3D hybrid simulator for large-scale multicellular systemPLOS ONE

Dear Dr. Sheng,

Thank you for submitting your manuscript to PLOS ONE. After careful consideration, we feel that it has merit but does not fully meet PLOS ONE’s publication criteria as it currently stands. Therefore, we invite you to submit a revised version of the manuscript that addresses the points raised during the review process.

The reviewers' comments highlighted that the manuscript is well-written and the results achieved are promising. However, there are a few concerns that the authors must address. A better description of the proposed parallelization is required, including GPU settings and library versions. A comparison between the accuracy of the proposed approach and state-of-art strategies is needed. Please refer to all the reviewers' reports for detailed comments which can help improve the current version of the manuscript

We look forward to receiving your revised manuscript.

Kind regards,

Andrea Tangherloni

Academic Editor

PLOS ONE

Journal Requirements:

Reviewers' comments:

Reviewer's Responses to Questions

**Comments to the Author**

1. Is the manuscript technically sound, and do the data support the conclusions?

Reviewer #1: Yes

Reviewer #2: Yes

2. Has the statistical analysis been performed appropriately and rigorously? 

Reviewer #1: N/A

Reviewer #2: N/A

3. Have the authors made all data underlying the findings in their manuscript fully available?

Reviewer #1: Yes

Reviewer #2: Yes

4. Is the manuscript presented in an intelligible fashion and written in standard English?

Reviewer #1: Yes

Reviewer #2: Yes

5. Review Comments to the Author

Reviewer #1: The article concerns GPU-supported 3D simulations of the multicellular dynamics. Such simulations are very desired in many domains like medicine, biology, chemistry, etc. However, they are very time consuming. There were many attempts to speed up such simulations using different platforms: computer clusters, GPUs, etc. The authors have referenced them in the introduction section. In comparison to them, in this manuscript, a new GPU-based simulator is proposed (probably faster and at least as accurate as previous ones).

The solution is interesting and seems to be fast as well as provides accurate results. It is well described from the mathematical and biological perspective. However, the title and strength of the solution is ‘GPU-powered…”. Thus, the introduced improvements/parallelization should be described in more detail, e.g., figures and maybe pseudocode. Without this, the reader would have to scan the source code.

Below the summary of the remarks:

1. The solution needs more description from the GPU-parallelization perspective.

2. What was the CUDA version?

3. There is no information about Gell simulations on CPU? time comparison with at least one-threaded version would be desired.

4. There is no information about GPU settings? And no information about their influence on the time and memory performance, what kind of memory does the GPU code use (shared, etc.)?

5. More GPU-support profiling would be desired.

6. Where are the conclusions?

7. Concerning Fig.8 and this case study: How the memory consumption was changed?

8. How the measurements in Table 5 were collected?

9. The authors have compared the proposed solution directly with one of CPU-parallelized simulators, why not with one of the previous GPU-supported ones?

10. Equation 1, what does r_a means ?

Reviewer #2: In this work the authors propose Gell, a GPU based approach to simulate large-scale systems on a cellular level. The results prove Gell efficiency in terms of speed-up, paving the way to novel and more in-depth analyses.

The paper is written in a good English and I enjoyed reading it.

However, there are many minors that need to be addressed before publication and one personal concern that I would like to see explicitly stated in the paper.

In the abstract the authors say "a fast and memory-efficient open-source GPU-based hybrid computational modeling platform Gell", I honestly had to read it a couple of times to understand that Gell was the name of the platform. I suggest them to "reverse" the sentence in something like "we develop Gell (GPU Cell), a fast and memory-efficient ...".

I think that this is a problem which will be addressed during the proof-reading process. Therefore I do not think the authors should worry about it now, but I prefer highlighting this problem: the bibliographic references are indicated with round brackets and not with squared brackets when squared brackets are usually used for the references.

In line 86, the authors use the term "cluster computers", should not "computer cluster" be a better term?

In line 89, I don't think that "shown" is the proper verb tense for the sentence. Also in line 94 the verb tense of "runs" should not be the proper one.

Table 1 is not referenced in the text.

In line 190, it is not clear what is the v_i with the dot above, the authors should mention it during the explaination of the equation symbols.

Moreover, the authors, should terminate each Equation with a dot or a comma and start the following sentence accordingly.

In line 194, the authors state "is like fluid drag", I suggest them to use another term that is not "is like".

In line 229, a column is missing to properly introduce the equation.

In lines 260 and 261, a reference to the Thomes algorithm is required.

I suggest the authors to extend and improve the overall quality of the "Cell cycle update" Section, since it is very short and I have the feeling that some potentially interesting technical details are not fully explained.

Figure 1 is not referenced in the text.

In line 356 and following, I suggest the authors to avoid use and introduce acronyms in the sub-section titles and introduce them in the sub-section text.

In line 364 a $\\mu$m should be used instead of um, if I properly understood the unit of measure.

Also in line 393,438 there is the same problem in terms of unit of measure, I suggest the authors to check all the unit of measure notations and fix them accordingly (such as also in lines 465 and 466).

I would also like to suggest the authors to use the explicit term to refer to Tables and Figures and not simply using "Tab" and "Fig".

The specific of the personal computer listed in line 271 should be moved at the beginning of the Results Section (line 355).

In line 416, "in vitro" should be attached as "in-vitro" and it should be in italic, since it is a latinism.

In Figure 6.c I would clearly point out that the x-axis is in logarithmic scale.

In Figures 6.a and 6.b, I suggest the authors to reduce and move the x10^6 from the x label towards a more "tick-level" location.

In line 455, the acronym is used in the title: as suggested before I kindly ask to move the acronym inside the section text. Moreover there is a typo in "carcinima" which should be spelled in "carcinoma".

In lines 481 and 482, I think that the Table reference is wrong, shouldn't it be to Table 5?

In line 520, also the "in vivo" term should be in italic.

I renew for the x label of Figure 8 what I suggested before for Figure 6.

In line 553, the term "the" does not properly fit in my opinion, I suggest the authors to use "these" or something similar.

The only concern I have about this work is relative to the mathematical stability of Gell. In the work, the authors compare Gell with the state-of-the-art in terms of execution speed and speed-up, but they never claim or prove that the output of both models are the same. My concern is raised from the fact that typically GPUs work on a 32-bit precision while floating numbers on CPUs work on a 64-bit precision level. I would kindly ask the authors if they can check that the simulations performed for both the softwares (i.e., the simulations used to compute the speed-up) produce the same output. The authors could explicitly state that the results are the same, if this is actually the case as I expect.

6. PLOS authors have the option to publish the peer review history of their article (what does this mean?). If published, this will include your full peer review and any attached files.

Reviewer #1: No

Reviewer #2: **Yes: **Daniele Maria Papetti

---

## [Author Response · Author response to Decision Letter 0]

26 May 2023

We responded all comments and questions in the document named "Gell_Response to Reviewers".

---

## [Decision Letter · Decision Letter 1]

4 Jul 2023

Gell: A GPU-powered 3D hybrid simulator for large-scale multicellular system

PONE-D-23-02925R1

Dear Dr. Sheng,

We’re pleased to inform you that your manuscript has been judged scientifically suitable for publication and will be formally accepted for publication once it meets all outstanding technical requirements.

Kind regards,

Andrea Tangherloni

Academic Editor

PLOS ONE

Additional Editor Comments (optional):

Reviewers' comments:

Reviewer's Responses to Questions

**Comments to the Author**

1. If the authors have adequately addressed your comments raised in a previous round of review and you feel that this manuscript is now acceptable for publication, you may indicate that here to bypass the “Comments to the Author” section, enter your conflict of interest statement in the “Confidential to Editor” section, and submit your "Accept" recommendation.

Reviewer #1: All comments have been addressed

Reviewer #2: All comments have been addressed

2. Is the manuscript technically sound, and do the data support the conclusions?

Reviewer #1: Yes

Reviewer #2: Yes

3. Has the statistical analysis been performed appropriately and rigorously? 

Reviewer #1: N/A

Reviewer #2: Yes

4. Have the authors made all data underlying the findings in their manuscript fully available?

Reviewer #1: Yes

Reviewer #2: Yes

5. Is the manuscript presented in an intelligible fashion and written in standard English?

Reviewer #1: Yes

Reviewer #2: Yes

6. Review Comments to the Author

Reviewer #1: No more comments, all remarks were addressed, ......................................................

Reviewer #2: I thank the authors to answer to my concerns.

I endorse the publication of the manuscript.

Regarding the line 299, the ":" symbol should be added after the closed squared brackt before Equation 11; i.e., "can be written as [24]:"

I also would like to thank the authors for the complete and clear answer to my last question.

7. PLOS authors have the option to publish the peer review history of their article (what does this mean?). If published, this will include your full peer review and any attached files.

Reviewer #1: No

Reviewer #2: No

---

## [Editor Report · Acceptance letter]

7 Jul 2023

PONE-D-23-02925R1 

Gell: A GPU-powered 3D hybrid simulator for large-scale multicellular system 

Dear Dr. Sheng:

I'm pleased to inform you that your manuscript has been deemed suitable for publication in PLOS ONE. Congratulations! Your manuscript is now with our production department. 

Kind regards, 

on behalf of

Dr. Andrea Tangherloni 

Academic Editor

PLOS ONE